# Small GTPases and Their Role in Vascular Disease

**DOI:** 10.3390/ijms20040917

**Published:** 2019-02-20

**Authors:** Alison Flentje, Richa Kalsi, Thomas S. Monahan

**Affiliations:** Division of Vascular Surgery, Department of Surgery, University of Maryland School of Medicine, 22 South Greene Street, Suite S10B00, Baltimore, MD 21201, USA; AFlentje@som.umaryland.edu (A.F.); Richa.kalsi@som.umaryland.edu (R.K.)

**Keywords:** small GTPases, cdc42, rac1, RhoA, vascular endothelium, restenosis, migration

## Abstract

Over eighty million people in the United States have cardiovascular disease that can affect the heart causing myocardial infarction; the carotid arteries causing stroke; and the lower extremities leading to amputation. The treatment for end-stage cardiovascular disease is surgical—either endovascular therapy with balloons and stents—or open reconstruction to reestablish blood flow. All interventions damage or destroy the protective inner lining of the blood vessel—the endothelium. An intact endothelium is essential to provide a protective; antithrombotic lining of a blood vessel. Currently; there are no agents used in the clinical setting that promote reendothelialization. This process requires migration of endothelial cells to the denuded vessel; proliferation of endothelial cells on the denuded vessel surface; and the reconstitution of the tight adherence junctions responsible for the formation of an impermeable surface. These processes are all regulated in part and are dependent on small GTPases. As important as the small GTPases are for reendothelialization, dysregulation of these molecules can result in various vascular pathologies including aneurysm formation, atherosclerosis, diabetes, angiogenesis, and hypertension. A better understanding of the role of small GTPases in endothelial cell migration is essential to the development for novel agents to treat vascular disease.

## 1. Introduction

Over 80 million people in the United States have cardiovascular disease, resulting in over seven million revascularization procedures each year. As the population ages and the prevalence of risk factors, such as obesity and diabetes increases, the number of patients whom will require revascularization procedures will rise [1]. Both endovascular and open surgical revascularization procedures are limited in durability by intimal hyperplasia. This occurs when surgical, balloon, or stent-associated trauma to the endothelium and underlying vessel wall initiates a sequence of events resulting in the dedifferentiation of medial vascular smooth muscle cells with migration to and subsequent proliferation in the intima. This results in a thickened, proliferative neointima that reduces the inner vessel diameter, eventually reducing blood flow to downstream tissues (Figure 1A) [2]. Formation of intimal hyperplasia significantly limits the durability of surgical and catheter-based revascularization procedures that often require surgical intervention (Figure 1B) or cause complications from a failed intervention, including myocardial infarction, stroke, or amputation.

There is currently no treatment to prevent the formation of intimal hyperplasia after bypass surgery or endarterectomy. A variety of techniques have exhibited variable and generally limited success to prevent the formation of intimal hyperplasia after surgery, angioplasty, or stenting including gene therapy to conduits [3,4], drug-coated balloons (paclitaxel) [5], drug-eluting stents (calcineurin inhibitors) [6,7], and radiation [8,9]. Despite an early enthusiastic response with drug-eluting stents, long-term outcomes might not differ significantly from bare metal stents [10]. All of these modalities inhibit both vascular smooth muscle cell and endothelial cell proliferation. Additionally, drug-eluting stents actually have an increased risk of late thrombosis (>1 year) compared to bare metal stents, which is likely due to incomplete reendothelialization, chronic inflammation, fibrin deposition, and platelet activation [11,12]. The threat of late thrombosis has led to extended-duration dual antiplatelet therapy, which carries significant risk of bleeding. While pathologic smooth muscle cell proliferation and migration contribute to intimal hyperplasia, prompt restoration, and preservation of an intact endothelium is essential for prevention of thrombosis and vessel patency. To date, there are few targets for cell type-specific regulation of proliferation [13]. There remains a large unmet clinical need for an agent that selectively inhibits vascular smooth muscle cells proliferation while accelerating the reconstitution of an intact endothelium. The reconstitution of an intact endothelium requires migration of endothelial cells to the area of injury, their subsequent proliferation, and reconstitution of adherens junctions. Endothelial cells are neither migratory nor proliferative in health. Augmenting endothelial cell migration and proliferation would be beneficial in a specific location (site of intervention) and period time (periprocedural) until the iatrogenic injury has resolved.

The central role of reendothelialization to create an intact vessel underscores the tremendous importance of understanding the mechanisms of regulation of endothelial cell migration. Currently, there is no clinical intervention to augment either the proliferation of endothelial cells in the recently injured artery, or more importantly, that potentiates endothelial cell migration into the denuded vessel. We have been able to dramatically increase the rate of reendothelialization in murine models of arterial injury by inhibiting the myristoylated alanine-rich C kinase substrate (MARCKS) signaling pathway (Figure 2) [14]. MARCKS signaling is essential for the regulation of small GTPase signaling in vascular smooth muscle cell migration [15]. The small GTPases play an essential role in cell migration. A better understanding of the molecular events driving endothelial cell migration is needed to develop new and innovative strategies to promote reendothelialization after vascular interventions.

The vascular endothelium is a selectively permeable membrane that lines the inner surface of the entire circulatory system. Although once believed to be a simple barrier [16,17], these cells play an enormous role in a variety of complex functions that impact vascular biology including nutrient exchange, homeostasis, blood vessel tone, and the immune system. In particular the interactions between endothelial cells and vascular smooth muscle cells play an important role in cardiovascular physiology and pathology [18]. The endothelium receives stimuli through the presence of membrane bound receptors and cell–cell connections that affect a diverse group of downstream effectors. The focus of this manuscript is one class of these effectors, the small GTPases. 

Endothelial cells contain mechanosensing molecules that detect the mechanical forces from blood flow and transduce it into intracellular biochemical responses. Shear stress is defined as the tangential frictional force acting on the surface of the endothelial cell [19]. This stress leads to an activation of various signaling cascades such as the opening of K^+^ and Ca^++^ channels, production of NO, release of ROS, tyrosine phosphorylation of proteins (Shc and c-src), activation of heterotrimeric G protein, and activation of kinases (focal adhesion kinase, mitogen-activated protein kinase, protein kinase C, and JNK) or nuclear transcription factors (c-fos, c-jun, c-myc, and NF-κB) that leads to increased gene expression of *intercellular cell adhesion molecule 1* (*ICAM-1*), *NOS*, *platelet derived growth factor* (*PDGF*), monocyte chemoattractant protein, and transforming growth factor (TGF-β) [20]. It is essential to the regulation of vascular remodeling, control of blood pressure, embryogenesis, and atherogenesis [20]. In part, this is mediated by the hemodynamic forces affecting the expression of adhesion proteins on the endothelium [20]. 

Endothelial cell response to shear stress is seen by elongation in the direction of flow and rearrangement of their microtubules [20]. Clinically this is seen in the distribution of atherosclerotic plaques that appear more frequently at areas of nonlaminar flow such as vessel branch points, bifurcations, and regions with high curvature [20]. Shear stress effects endothelial cell permeability in a stress-dependent manner where increased permeability occurs within sites of low shear stress [21]. These sites have greater macromolecule accumulation such as LDL, leading to atherosclerotic lesions [22]. Endothelial dysfunction leading to increased vascular permeability to LDLs is increased by smoking and high fat diet linking atherosclerosis to lifestyle [23,24]. 

Endothelial cell migration, proliferation, and barrier function are essential to restore blood vessel health after injury. The small GTPases drive these cellular processes and consequently the regulation of their function are an attractive target for therapy. However, these molecules are critical regulators of cellular processes in many cell types. The small GTPases are implicated in multiple vascular pathologies in addition to reendothelialization, both over expression and silencing of their function have salutary and detrimental effects depending on the affected cell type. 

## 2. Overview of the Small GTPases

The small GTPases are major regulators of diverse mechanobiology. These enzymes function as a molecular switch controlled by guanine nucleotide exchange factors (GEFs) that catalyze the exchange of guanosine diphosphate (GDP) to guanosine triphosphate (GTP). When the enzyme is phosphorylated it is in its active form and it is inactivated by GTPase-activating proteins (GAPs) that stimulate GTP hydrolysis [25]. Some of these molecules are maintained in their inactive form through interactions with guanosine nucleotide dissociation inhibitors (GDIs) (Figure 3) [26]. Most of small GTPases contain a lipid modification to the C-terminus that localizes them to the cell membrane. GDIs can act by binding to this lipid group and thus extracting them from the membrane [27]. Upstream kinases, including Src, protein kinase C (PKC), and protein kinase A (PKA), target GDIs and modulate their affinity to the Rho GTPases [28].

The 20 members of Rho GTPase family are divided into six subfamilies: Rho, Rac, Cdc42, Rnd, RhoBTB, and RhoT/Miro. Of these, Rho, Rac, and cdc42 have been studied the most in relation to vascular disease and cancer biology and are described as the classic Rho GTPases that function through the opposing actions of GEFs and GAPs as described above. This is in contrast to the atypical small GTPase like Rnd and RHOU that are unable to hydrolyze GTP and thus are constitutively GTP-bound [27]. This review will focus on the classic Rho GTPases—Rho, Rac, and Cdc42.

The Rho GTPase family is a key regulator of actin reorganization and thus cell migration. In cell migration the first step is the rapid polymerization of actin to form stress fibers at the leading edge of a cell. Stress fibers are contractile bundles in non-muscle cells composed of actin and myosin. In migration they form protrusive structures. The simplest of these structures are filopodia, which are slender cylinder cytoplasmic projections. More complex protrusion sheets composed of a mesh of actin that dominant the leading edge is called lamellipodia. Lamellipodia form a characteristic ruffle as the mesh lifts up off the substrate and moves backwards during locomotion [29]. Rho is essential to both the formation of focal adhesion and actin stress fiber assembly [30]. Rho activates the downstream target Rho-associated kinase (ROCK) that phosphorylates myosin light chain phosphatase (MLC) and LIM kinase (LIMK), which contribute to direct actin contractility and the first step of actin polymerization, actin nucleation via phosphoregulation of cofilin. Rho also affects formin proteins via mDIA which allows for actin polymerization and microtubule stabilization. Rac1 regulates the formation of membrane ruffles, adherent junctions, and lamellipodia [31], while Cdc42 is required for the formation of filopodia [32]. Both Rac1 and Cdc42 modulate motility via similar mechanisms. LIMK is activated by Rac1 and Cdc42 via p21 activated kinase1 (PAK1). Rac1 and Cdc42 both activate the Wiskott–Aldrich syndrome family of proteins that include WASP, N-WASP, and WAVE. These proteins activate Arp2/3 promoting actin polymerization [28]. 

Restoration and maintenance of an intact endothelium is essential for vessel patency and vascular function. The small GTPases are key regulators of endothelial cell migration and adherence junctions which regulate endothelial permeability. However, the small GTPases are also associated with vascular pathologies including aneurysm formation, atherosclerosis, diabetes, angiogenesis, and hypertension. A better understanding of the role of small GTPases in endothelial cell migration is essential to the development for novel agents to treat vascular disease.

### 2.1. Ras Homolog Gene Family, Member A (RhoA)

The Ras homolog gene family, member A (RhoA) was the first member of the Rho GTPase superfamily to be discovered in 1985 by Richard Axel [33]. RhoA was originally studied in cancer cells where it was found to stimulate cell cycle progression and migration [34]. In the last two decades the RhoA subfamily has been identified as an important regulator of endothelial function and implicated in cardiovascular disease. 

An important downstream target of Rho is the Rho-associated protein kinase (ROCK). These ubiquitously expressed serine/threonine protein kinases are involved in diverse cellular activities including apoptosis, smooth muscle contraction, motility, cell adhesion, actin cytoskeleton assembly, and remodeling of the extracellular matrix [35]. In regulation of endothelial cell migration ROCK interacts with ezrin, radixin, and moesin also known as the ERM proteins that function as cross-linkers between the plasma membrane and actin filaments. The ERM proteins have also been shown to coordinate the localization of leukocyte adhesion molecules that is critical to barrier function [36]. The best-known substrates of ROCK is myosin phosphatase target subunit 1 (MYPT1), which increases MLC phosphorylation leading to actin contraction [37]. As discussed, earlier LIMK is another important substrate of ROCK that inhibits actin-depolymerization and thus stabilizes actin filaments [38]. 

RhoA inhibition has been shown to prevent vascular endothelium growth factor (VEGF)-enhanced endothelial cell migration in response to vascular injury but has no effect on basal endothelial cell migration [39]. Mastumoto et al. targeted the RhoA/ROCK pathway using fasudil—a ROCK inhibitor—in a large animal coronary stent model. Fasudil reduced coronary artery in-stent restenosis by significantly reducing inflammation and collagen deposition [40]. Fasudil has not been used clinically to prevent restenosis, but these findings highlight the great potential for antagonism of the RhoA/ROCK pathway A for the treatment of vascular proliferative disease. 

A critical function of endothelial cells is the maintenance of the endothelial barrier. Similar to Cdc42, RhoA is a critical mediator of the integrity of adherens junctions. Szulcek et al. described different subcellular locations of RhoA and its importance in maintaining endothelial barrier integrity [41]. This included an increase in RhoA activity at cell margins prior to the closure of transient spontaneous gaps between nonstimulated endothelial cells and a thrombin-mediated disruption or enhancement of integrity depending on cellular location [41]. This finding has been further supported by the work of Amerongen et al. who demonstrated basal activity of Rho contributes to barrier integrity by regulating VE-cadherins but also contributes to dysfunction by inducing contractility of F-actin [42]. These data demonstrate that RhoA plays a role in not only barrier integrity but also dysfunction based on subcellular location and level of activity. 

At sites of increased permeability there is cytoskeletal remodeling regulated by Rho and Rac. In a thrombin induced model of lung injury Rho is rapidly activated by p115RhoGEF leading to translocation to the cell membrane, which triggers MLC phosphorylation, stress fiber formation, and actinomycin contraction thus causing increased permeability [43]. Under static conditions, both Rho and Cdc42 are distributed throughout the cytosol, but under shear stress conditions they localize to the plasma membrane. Cdc42 was found to independently activate the AP1/12-*O*-tetradecanoyl-13-phorbol-acetate–responsive element (TRE) in response to JNK induced by shear stress, while Rho activates ROCK leading to the actin-based reorganization of the cytoskeleton [44]. In diabetes advanced glycation end products (AGEs) accumulate in the vasculature triggering a series of functional and morphologic changes of endothelial cells. In particular Wang et al. demonstrated that increased AGEs cause a dose dependent increase in the RhoA/ROCK pathway activation leading to increased endothelial cell permeability that can cause a multitude of diabetic complications [45]. Sun et al. further demonstrated that RhoA/ROCK mediates VEGF induced microvascular endothelial hyperpermeability [46]. 

The RhoA/ROCK pathway also plays a role in remodeling and repair as a key determinant for the lineage fate of mesenchymal stem cells (MSCs). One predominant theory for the formation of intimal hyperplasia is that nestin-positive MSCs mobilize to the site of vascular injury. TGF-β-activated RhoA causes the MSCs to primarily differentiate into neointima SMC/myofibroblast cells. When RhoA is inhibited the MSCs secrete matrix metalloproteinase-3 (MMP3) which degrades connective tissue growth factor (CTGF) and causes release of VEGF promoting differentiate into endothelial cells for repair [47]. Controlling the fate of MSCs through RhoA/ROCK antagonism represents a potential novel therapy to prevent the formation of intimal hyperplasia.

Regulation of endothelial nitrous oxide synthase (eNOS) is an important role for ROCK/RhoA in the vascular endothelium. Nitric oxide (NO) is a potent vasodilator produced by eNOS in the endothelium and is the principal regulator of vascular tone in mammals. NO-induced vasodilation occurs via activation of myosin light chair phosphatase (MLCP) in a cGMP dependent manner. RhoA/ROCK counteracts this through MLCP inactivation and calcium desensitization. Hyperglycemia causes diabetic induced vascular dysfunction due to inhibition of eNOS and thus less endothelial NO production secondary to the upregulation of ROCK [48]. Statins affect the post-translational modification of RhoA, called geranylgeranylation, leading to a decrease in membrane translocation and GTP binding activity. Thus, eNOS expression is upregulated by improving mRNA stability [49]. ROCK/Rho decreases eNOS expression and affects the availability of nitric oxide (NO), which contributes to cardiovascular pathology through the regulation of vascular tone, platelet aggregation, smooth muscle cell proliferation, and leukocyte recruitment to the vessel wall [50].

Due to its critical role in endothelial barrier function, migration, and eNOS activity, the Rho pathways have been identified as a possible target for therapeutic intervention. Anagliptin, a dipeptidyl peptidase IV inhibitor, is a new anti-hyperglycemic medication being used in the treatment of Type 2 diabetes. Pretreatment with this novel drug prior to wire injury has been found to reduce intimal hyperplasia and stimulates endothelial cell migration by upregulating the level of superoxide dismutase (SOD-1), which reversed the downregulation of RhoA expression and c-Jun N-terminal kinase (JNK) signaling [51]. 

In summary, RhoA is a central regulator of critical cellular functions in vascular tissue. Its activity drives the polymerization and depolymerization of actin fibers directing endothelial cell migration. In addition to directly interacting with the cytoskeleton, it regulates NO synthesis, directs the differentiation of MSCs, regulates endothelial permeability, and collagen deposition at the site of injury. RhoA/ROCK inhibitors have been used in animal models of vascular injury to prevent restenosis. However, more study is needed to better understand the mechanism of action specific to endothelial and smooth muscle cells as RhoA/ROCK antagonism would have both salutary and harmful effects on the different cellular processes leading to intimal hyperplasia formation.

### 2.2. Cell Division Control Protein Homolog 42 (Cdc42)

Cell division control protein homolog 42 (Cdc42) is the most studied member of Cdc42 subfamily due to its role in the pathogenesis of cancer, neurodegenerative disease, and cardiovascular disease [52]. Although relatively rare, mutations in Cdc42 have been associated with structural cardiac malformations, immune, hematologic, lymphatic, and neurologic abnormalities. Currently there is no described human mutation specifically associated with peripheral blood vessel formation or function, which underscores the importance of Cdc42 function in many cell types [53].

The importance of Cdc42 in embryonic development has been well described in the literature. Mice with endothelial specific knockout of Cdc42 causes embryonic lethality due to defects in angiogenesis and increased microvascular permeability [54]. This finding is supported by evidence that Cdc42 activity was critical to regulating and maintaining cell–cell connections in angiogenesis. Cdc42 engages actin fiber to VE-cadherin that stabilize junctional complexes during endothelial migration in multicellular sprouts [55]. When endothelial cell specific deletion of Cdc42 is induced, there is an increase in the fragmentation of vascular endothelium growth receptor 2 through DAM metallopeptidase domain 17 (ADAM17) causing defects in migration of endothelial cell but no change in proliferation [56]. Similar defects in migration are observed with Cdc42 deletion in other cell lines. Cdc42 deletion results in severe defects in renal epithelial cell adhesion, polarization, and migration. The total of these defects prevents uteric bud and mesenchyme development resulting in renal failure [57].

Besides playing a key role in the development of blood vessels, Cdc42 has also been found to play an important role in endothelial regeneration and vascular repair. Cdc42 knockdown leads to inhibition of migration of endothelial cell resulting in impaired vascular recovery and increased vascular permeability in an acute lung injury model [58]. In pulmonary endothelial cells Cdc42 regulates PAK1. PAK1 is a serine/threonine kinase effector of the Rho GTPases including Cdc42 and Rho that links the small GTPases to cytoskeletal organization. In endothelial cells, PAK1, acting as an effector for Cdc42, is essential for vascular remodeling; deficiency of Cdc42 prevents endothelial regeneration after vascular injury [58]. In addition to regulating endothelial cell regeneration, Cdc42 is an important regulator of endothelial cell barrier function. Vascular permeability is principally regulated by adherens junctions (AJs) between adjacent endothelial cells. These cells form an impermeable barrier at the AJ through their VE-cadherin connections. Kouklis et al. demonstrated the importance of Cdc42 in vascular recovery. Following a thrombin challenge, which is known to disassemble AJs, Cdc42 activation occurred at 1 h, which paralleled the reformations of AJs. At this time point, Cdc42 was shuttled from the cytosol to plasma membrane thus restoring endothelial junctional integrity [59].

Atherosclerosis is a process attributed to age related chronic inflammation and endothelial dysfunction. Cdc42 deletion in endothelial cells significantly attenuates chronic inflammation and plaque formation in atherosclerosis. This effect is caused by Cdc42-mediated activation of NF-κB and p-53 induction of proinflammatory genes in senescent endothelial cell [60]. This is supported by data that Cdc42 is upregulated in senescent endothelial cells through activation of the scaffolding protein caveolin-1 resulting in the characteristic morphologic changes through modulation of focal adhesion, actin stress fiber, and membrane ruffle formation [61]. Senescent endothelial cells accumulate in atherosclerotic lesions and show characteristic morphologic changes such as a flat large cell shape associated with functional decay and growth arrest. These morphologic changes could be attributed to Cdc42 activation and its role in plaque formation.

Our lab has demonstrated the upregulation of myristoylated alanine-rich C-kinase substrate (MARCKS) plays an important role in the formation of intimal hyperplasia through increased availability of membrane-associated phosphatidylinositol 4,5-bisphosphate (PIP_2_) levels leading to Cdc42 and Rac1 activation in vascular smooth muscle cells [15]. A similar relationship between the regulation of Cdc42 and membrane availability of PIP_2_ by MARCKS has been demonstrated in developing neurites [62]. The MARCKS family of protein is a membrane associated with a 25 amino acid membrane domain that modulates actin filament assemble. The MARCKS effector domain contains the acidic lipid, PIP_2_ which can be inhibited by the binding to calcium/calmodulin or due to the phosphorylation by PKC [37]. We are currently investigating the role of MARCKS and Cdc42 regulation in endothelial cell migration.

In summary, Cdc42 is an essential regulator of both endothelial cell migration and barrier function. However, Cdc42 activation also plays a role in the formation of atherosclerosis through activation of proinflammatory genes in endothelial cells. This could be in part due its upregulation in senescent endothelial cell that cause morphologic and functional changes leading to the accumulation of endothelial cells in plaques. Even though it has a role in vascular pathology, it is also essential to vascular homeostasis. Cdc42 activity is essential to angiogenesis, cell migration, endothelial barrier function, and vascular recovery via the reformation of AJ.

### 2.3. Ras-Related C3 Botulinum Toxin Substrate 1 (Rac1)

The ras-related C3 botulinum toxin substrate (Rac1) is another member of the Rho superfamily of small GTPases that was first described in 1989 [63]. Early work with Rac1 demonstrated its importance in cell motility and in the metastatic process. Overexpression of Rac1 in mice results in tumor formation closely resembling human Kaposi’s sarcoma [64]. Rac1 is now established as a key regulator of endothelial cell motility and is integrally involved in directing vascular development. While Rac is upregulated in many clinical forms of cancer. Mutations are rarely associated with specific disease states. However, a known Rac1 mutation—RAC1^P29S^—is the third most common mutation associated with melanoma. Currently, work is being done to develop specific Rac1 inhibitors for the treatment of melanoma [65,66].

Angiogenesis or the sprouting of new blood vessels is initiated by VEGF induced migration of endothelial cells via induction of lamellipodia. VEGF activates Rac1 through the VEGFR-2. Rac1 is thus essential to VEGF-induced endothelial cell migration [67]. This is supported by data that when Rac1 is deleted in endothelial cells it results in embryonic lethality with resultant defects in the development of major vessels and complete lack of small branched vessels in the embryos and their yolk sack [68].

The mechanosensory stimulus caused by blood flow involved in shear stress initiates several pathways including the small GTPase Rac1. Liu et al. demonstrated that platelet endothelial cell adhesion molecule-1 (PECAM-1) is required for GTP loading of Rac1 via the activation of Src, which phosphorylates and thus activates the Rac GEF, Vav2 [69]. Additionally, vascular endothelial cadherin (VE-cadherin) is essential to the polarized activation of Rac1 in response to sheer stress via the formation of a VE-cadherin-Par3-Tiam1 complex, which allows coupling of p67phos subunit of NADH oxidase [69]. Thus, when VE-cadherin, Para3, or Tiam1 is depleted in endothelial cells there is impaired ROS production in response to shear stress [67]. In the response to shear stress Rac1 contributes to ROS production via NF-κB dependent increase in *ICAM-1* gene expression [70]. In an unstimulated cell NF-κB is sequestered to the cytoplasm and upon activation IκB is phosphorylated and degraded allowing NF-κB to translocate to the nucleus and bind to promoters which increase leukocyte adhesion and transendothelial migration to the atherosclerotic plaque [70]. 

Rac1 additional affects migration in response to shear stress. Huang et al. demonstrated that shear stress can induce Rac1 expression and promote cell migration while Rac1 mutation inhibited migration [71]. Further, Hu et al. demonstrated that transfecting cells with a dominant negative mutant of Rac1 prevented lamellipodia protrusion and cell migration in both static and shear tissue culture models while transfecting cells with an activated mutant induced lamellipodia but attenuated the shear-induced migration [72]. This indicates that Rac1 is essential at an appropriate level for cell polarization and shear stress induced migration. Rac1 has also been found to stabilize microvascular endothelial barrier functions likely due to increased junction association with the actin cytoskeleton [73]. Knezevic et al. found that T-lymphoma invasion and metastasis protein 1 (Tiam1) was required for Rac1 activation and localized actin polymerization [74]. This is supported by Walsh et al. that found Tiam/Rac1 causes endothelial barrier stabilization in response to shear stress [75].

Statins are 3-hydroxy-3-methyl-glutaryl-CoA (HMG-CoA) inhibitors that have become the standard in treatment of hypercholesterolemia and prevention of cardiovascular disease. Statins have been shown to promote the phosphorylation of AMP-activated protein kinase (AMPK) that activates Rac1. Simvastatin has also been shown to induce an approximately 2-fold increase in endothelial cell migration. This simvastatin-induced migration was completely attenuated by siRNA-mediated knockdown of Rac1 or AMPK [76]. In addition to potentiating the effects of Rac1 on endothelial cell migration, statins combat the proinflammatory phenotype generated by the over expression of ROS. Statins have a pleiotropic therapeutic effect by reducing the low density lipoprotein (LDL) cholesterol through the inhibition of HMG-CoA reductase and increasing small GTP-binding protein GDP dissociation stimulator (SmgGDS) that binds Rac1 targeting it for nuclear proteasome degradation [77]. The reduction in reactive oxygen species (ROS) would improve endothelial cell function and reduce pathology. 

ROS are reactive chemical mediators in which an oxygen molecule gains an electron. ROS are produced as a byproduct of cellular stressors and increased ROS levels have been associated with human pathologies including atherosclerosis, hypertension, renal dysfunction, and AAA formation. With regards to human pathological states, the association between chronic inflammation ROS and aneurysm formation has been well studied [78,79]. The causative mechanism between ROS and AAA formation has been further studied in mouse models of AAA formation. Administration of NADPH oxidase, apocynin, prevents aneurysm expansion. It is clear that modulating the NADPH oxidase pathway could be a valuable therapeutic intervention to prevent the vascular sequelae seen with increased ROS in particular AAA formation.

The generation of ROS by Rac1 activation of NADPH oxygenase (NOX) is well described in nonphagocytic vascular cells including fibroblasts [80], endothelial cells, and vascular smooth muscle cells [81]. Rac has several isoforms (Rac1, Rac1b, Rac2, and Rac3), but Rac1 is ubiquitously expressed and likely the main isoform contributing to NOX activation in nonhematopoietic cells [82]. Rac1 is critical to NOX assembly [83]. In endothelial cells Rac1 dependent NOX activation plays an important role in atherosclerosis and aneurysm formation via a number of vasoactive peptide hormones/steroids like angiotensin-II and aldosterone. Aldosterone induces angiotensin-converting enzyme (ACE) which generates angiotensin II (ATII) and subsequently activates angiotensin II receptor type 1 (AT1R) inducing NOX activation and thus the generation of ROS. Under physiologic conditions activation of this pathway increases blood pressure and when dysregulated can enhances vascular hypertrophy [84]. This pathway can act as a potential point of regulation in the generation of ROS for therapy to prevent vascular pathology. Marinkovic et al. demonstrated that using the Rac1 inhibitor azathioprine during ATII infusion in mice there is a reduction in aneurysm due to decrease in JNK phosphorylation in endothelial cells [85]. 

In endothelial cells ATIR activation by ATII also leads to the stimulation of PKC that phosphorylates MARCKS. Using FRET, Kalwa et al. demonstrated that Ang-II treatment leads to localized increases in PIP_2_ at cellular protrusion zones, which changes the activity of cytoskeleton associated Rho GTPases. In particular, siRNA-mediated knockdown of Rac1 negates the response of ATII and H_2_O_2_ on MARCKS. Further, the siRNA-mediated knockdown of Rac1, MARCKS, or cAbl leads to a hyperactive dysregulated cytoskeleton ultimately affecting cell structure and adhesion [86]. The commonly used ATR antagonist, Losartan, therefore blocks ATII induced phosphorylation of MARCKS. The Rac1/NOX pathway is also inhibited indirectly through the use of ACE inhibitors that are known to protect against vascular disease [87].

In summary Rac1 is essential for endothelial cell migration, specifically VEGF-induced angiogenesis that is critical to vascular development. Of all the Rho GTPases, Rac1 has the greatest potential for endothelial cell-specific therapy, as VEGF is a specific mitogen for ECs. As with other small GTPases, Rac1 demonstrates multiple effects depending on the cell type and conditions. Rac1 is required for the assembly of NOX but overexpression has been associated with aneurysm formation. This mechanism has been targeted by Rac1 inhibitors, statins, and ACEI to reduce vascular pathology. Statin therapy is known to decrease inflammation. However, here we described a mechanism through which statins potentiate endothelial cell migration through activation of Rac1, and another mechanism through which statins reduced inflammation by targeting Rac1 for degradation. 

## 3. Conclusions

In conclusion, the complications associated with long-standing cardiovascular disease continue to cause significant patient morbidity. The treatment with both endovascular and surgical revascularization procedures are limited in durability by intimal hyperplasia, which is attributed to endothelial dysfunction with pathologic smooth muscle cell proliferation and migration. As there is no intervention to augment the reendothelialization of an injured blood vessel, the molecular mechanism underlying the process needs to be explored to develop new and innovative surgical strategies.

Small GTPases are critical for key cellular processes such as migration and proliferation in many diverse cell types. Clearly, systemic inhibition of a small GTPase has the potential for myriad undesirable off-target effects. Development of a cell type-specific agonist or antagonist would have great therapeutic potential. Therapeutic manipulation of the small GTPases is particularly attractive for the treatment of surgical diseases. We have the ability to directly treat blood vessels through pressurized perfusion, as in the PREVENT trial [3], or with topical treatment of the target vessel, as in the PATENCY trial [88]. Our group has extensive experience delivering siRNA to isolated vessels and in vivo [14,89,90]. This approach would allow for a tissue specific, GTPase-specific inhibition of signaling. To further expand clinical uses for this technology, we are currently developing a controlled release siRNA-coated intravascular stent. Such a stent would allow us to locally deliver siRNA using minimally invasive vascular techniques.

The small GTPases are critical for regulating endothelial cell migration and AJ integrity. By functioning as a molecular switch these molecules act as major regulators of diverse mechanobiology. Specifically, Cdc42 activate proinflammatory genes in endothelial cells, but is also essential to angiogenesis, cell migration, endothelial barrier function, and vascular recovery via the reformation of AJ. Rac1 plays a key role in VEGF-induced angiogenesis that is critical to vascular development and NOX assembly. RhoA drives the polymerization and depolymerization of actin fibers directing endothelial cell migration and regulates NO synthesis, directs the differentiation of MSCs, determines endothelial permeability, and collagen deposition at the site of injury. In addition to regulating or enhancing endothelial cell migration and barrier function, GTPase activity is also essential for angiogenesis and the differentiation of MSCs to the endothelial lineage. However, activity of these molecules is also associated with vascular pathology including atherosclerosis, inflammation, and collagen deposition. The manipulation of these pathways is an attractive approach to enhance the reendothelialization at the time of vascular injury.

In summary, there is no agent used clinically to enhance reendothelialization. Key events in reendothelialization are endothelial cell migration and restoration of barrier function through adherence junctions. Both of these processes are highly regulated by the small GTPases making activation of these molecules an attractive target for therapy. However, because GTPases are essential to critical cellular functions in a wide variety of cells, overexpression has great potential for unintended consequences. More study is needed to understand the upstream and downstream messengers in these pathways and to identify cell type-specific agonists and antagonists of these pathways. One example is the VEGF-specific effect of Rac1 on endothelial cell migration. In addition to identifying cell type-specific signaling pathways, the development of endothelial cell-specific agonists and antagonists would greatly increase the clinical utility of this class of drugs. The small GTPase pathways have tremendous potential to fill the large unmet clinical need for therapy to enhance reendothelialization after vascular injury.

## Figures and Tables

**Figure 1 ijms-20-00917-f001:**
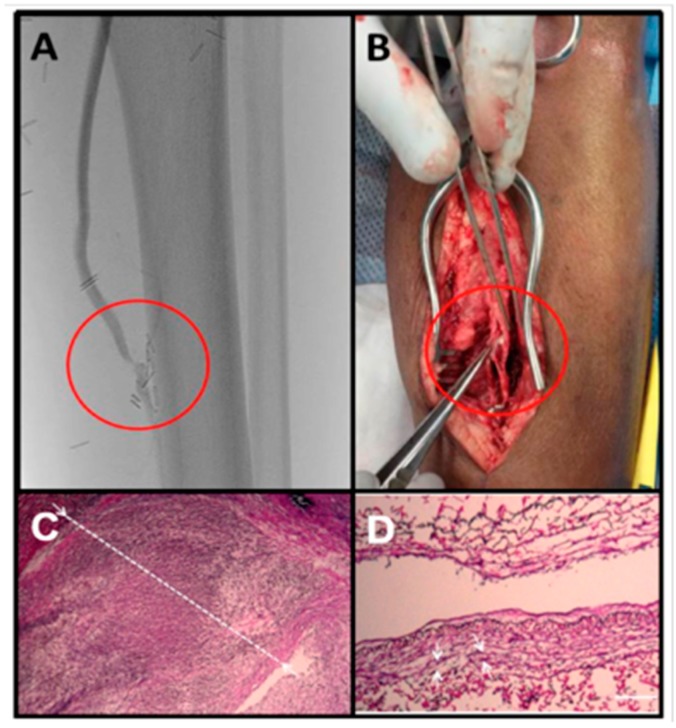
Intimal hyperplasia of a vein graft at the distal anastomosis. This bypass graft was failing based on surveillance duplex ultrasound examination. (**A**) Arteriogram confirmed a stenosis at the distal anastomosis. (**B**) This patient ultimately had an open surgical revision of this anastomosis. (**C**) Pathology demonstrated a dense cellular lesion with a greatly thickened intima (white line) and a nearly obliterated lumen. (**D**) An adjacent segment of relatively normal appearing vein graft is presented for comparison (Scale bar = 100 μm).

**Figure 2 ijms-20-00917-f002:**
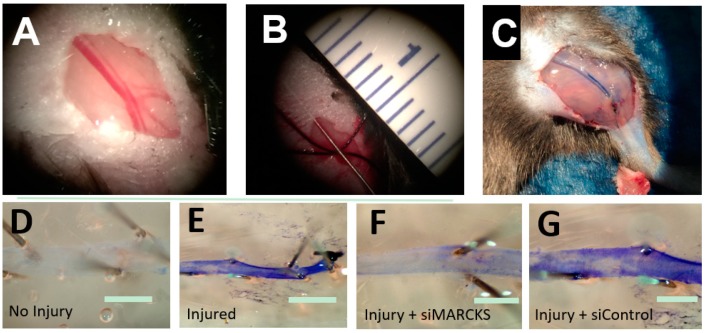
MARCKS knockdown decreases the time to reestablishment of an intact endothelium. (**A**) The mouse femoral artery (**B**) was denuded by passage of a 0.014 inch-diameter wire. (**C**) At the time of euthanasia, the animal was perfused with 0.3% Evans Blue dye which stained the damaged artery. (**D**) A normal, intact endothelium prevents Evans Blue from binding to the basement membrane. (**E**) Damage to the endothelium allows Evans Blue to bind to the vessel. (**F**) Vessels treated with siMARCKS reestablished baseline endothelial function three days after injury. (**G**) Nontargeting siControl exhibited little return of endothelial function at three days (scale bars = 100 μm).

**Figure 3 ijms-20-00917-f003:**
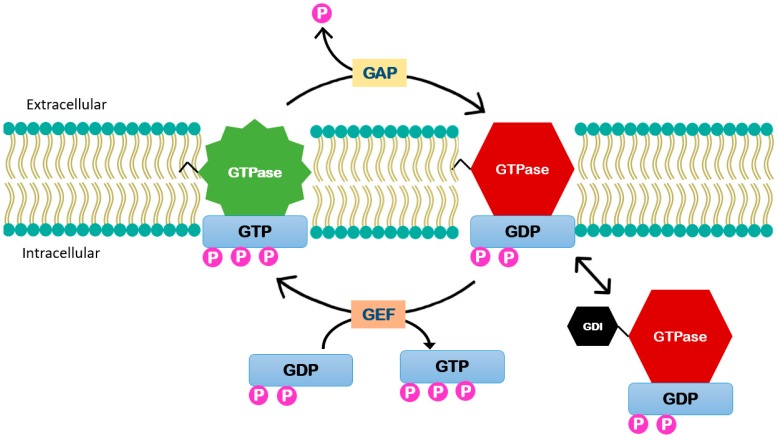
Small GTPases enzymes function. These enzymes function is controlled by (GEFs) that catalyze the exchange of GDP to guanosine triphosphate GTP activating the enzyme. It is inactivated by GTPase-activating proteins (GAPs) that stimulate GTP hydrolysis. Guanosine nucleotide dissociation inhibitors (GDIs) can act by binding to this lipid group and thus extracting them from the membrane.

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
