# Peer review of "Small GTPases and Their Role in Vascular Disease"

_ijms, 2019, doi:10.3390/ijms20040917_

Round 1

Reviewer 1 Report

Flentje et al have produced a comprehensive and detailed review of the role of small GTPases in vascular signalling and injury. This paper has a number of strengths that include a high level of information on various small GTPases. 

My only reasonable concern with this manuscript is with matching the content to the title

The Essential Role of Endothelial Cells and Small GTPases in the Treatment of Vascular Disease

The concept/topic of treatments targeting these molecules is only occasionally touched upon and I certainly did not finish this paper thinking that small GTPases have an essential role in vascular disease. 

The authors should emphasise this role more throughout the text.

While the review expertly compiles the current literature I feel that it lacks perspective from the authors- I would enjoy more insight onto where, based on the information they have presented, the field will go in this area in the future.

There are a number of small improvements to written English required particularly in the figure legends

Author Response

Flentje et al have produced a comprehensive and detailed review of the role of small GTPases in vascular signaling and injury. This paper has a number of strengths that include a high level of information on various small GTPases. 

My only reasonable concern with this manuscript is with matching the content to the title:

“The Essential Role of Endothelial Cells and Small GTPases in the Treatment of Vascular Disease”

We have changed the title of the manuscript to: “Small GTPases and Their Role in Vascular Disease.”

The concept/topic of treatments targeting these molecules is only occasionally touched upon and I certainly did not finish this paper thinking that small GTPases have an essential role in vascular disease. 

The authors should emphasize this role more throughout the text.

We have increased the emphasis on the potential therapeutic applications of modulating small GTPase activity and the challenges that we anticipate bring these concepts into a clinical model.  Specifically, we have added the second paragraph to the Conclusions section (lines 458-470). 

While the review expertly compiles the current literature, I feel that it lacks perspective from the authors- I would enjoy more insight onto where, based on the information they have presented, the field will go in this area in the future.

We have added our contributions to the literature.  We have modified the Specifically, these additions are the citations to references 14, 15, 90, 91.

There are a number of small improvements to written English required particularly in the figure legends.

We have revised the manuscript and have sought additional editorial assistance.

Reviewer 2 Report

This review article discussed three major small GTPases: Cdc42, RhoA, and Rac1 and their role in regulating cell proliferation, migration and physiological function in endothelial cells. The authors aimed at providing information that will ultimately used to facilitate re-endothelialization following revasculariztion procedures.

Because these GTPases have multiple effects to endothelial cell base on different conditions (and sometimes contradictory outcomes), the manuscript can be improved if the authors can provide discussion on the potential treatments for re-endothelialization by manipulating these GTPases base on the current knowledge. Providing such discussion can significantly increase the merit of this manuscript rather than stating that further study is needed.

Figure 1 is unnecessary because the phenomenon is well-documented. Also, the image quality is poor it provides little information to facilitate the discussion in the manuscript. 

Page 3, Line 83; Page 4, Line 144; Page 8, Lines 293, 296; Page 10, Line 399: the words "effect(s)" should be "affect(s)".

Page 3, Line 103: the word "manor" should be "manner".

Page 5, Lines 180-182: please add the reference for the following sentence: "Cdc42 knockdown lead to inhibition of migration of endothelial cell resulting in impaired vascular recovery and increased vascular permeability in an acute lung injury model."

Page 5, Line 194: do the authors mean "cytosol to plasma membrane"?

Page 6, Line 222: the word "through" should be "though".

Page 7, Lines 266-267: The sentence "Li et al. demonstrated that in response to shear stress induced translocation of Rho and Cdc42 from the cytosol into the plasma membrane." is confusing. Please perform a grammar check.

Page 7, Line 286: "nitrous oxide" should be "nitric oxide".

Page 8, Line 310: the word "more" should be removed.

Page 9, Line 341: "additional" should be "additionally".

Page 9, Line 344: the quotation mark should be removed.

Page 9, Line 346: "mutate" should be "mutant".

Page 9, Line 379: "Rac1 has several isoforms, but Rac1 is ubiquitously..." which Rac1 isoform?

Page 10, Line 393: "ATIR" should be "AT1R".

Author Response

This review article discussed three major small GTPases: Cdc42, RhoA, and Rac1 and their role in regulating cell proliferation, migration and physiological function in endothelial cells. The authors aimed at providing information that will ultimately be used to facilitate re-endothelialization following revasculariztion procedures.

Because these GTPases have multiple effects to endothelial cell base on different conditions (and sometimes contradictory outcomes), the manuscript can be improved if the authors can provide discussion on the potential treatments for re-endothelialization by manipulating these GTPases base on the current knowledge. Providing such discussion can significantly increase the merit of this manuscript rather than stating that further study is needed.

We have clarified how we hypothesize that each of the GTPases modulated to affect cardiovascular disease.  We have also included our own data regarding the in vivo effects of increasing the rate of re-establishment of an intact endothelium (reference 14).

Figure 1 is unnecessary because the phenomenon is well-documented. Also, the image quality is poor it provides little information to facilitate the discussion in the manuscript. 

We agree that intimal hyperplasia is a well-described pathologic process. However, most of the readership of this manuscript are not clinicians and are far less likely to be familiar with this pathology.  It serves a graphic illustration of the importance of the process in limiting the durability of arterial reconstruction procedures.  We have elected to keep the figure in the manuscript.  We have reformatted the figure from the original images .  

Page 3, Line 83; Page 4, Line 144; Page 8, Lines 293, 296; Page 10, Line 399: the words "effect(s)" should be "affect(s)".

We have made the changes to the manuscript.

Page 3, Line 103: the word "manor" should be "manner".

We have made the change to the manuscript.

Page 5, Lines 180-182: please add the reference for the following sentence: "Cdc42 knockdown lead to inhibition of migration of endothelial cell resulting in impaired vascular recovery and increased vascular permeability in an acute lung injury model."

The reference has been added.

Page 5, Line 194: do the authors mean "cytosol to plasma membrane"?

We did intend to say plasma membrane.  We have made the change to the manuscript.

Page 6, Line 222: the word "through" should be "though".

We have made the change to the manuscript.

Page 7, Lines 266-267: The sentence "Li et al. demonstrated that in response to shear stress induced translocation of Rho and Cdc42 from the cytosol into the plasma membrane." is confusing. Please perform a grammar check.

We have made the change to the manuscript.

Page 7, Line 286: "nitrous oxide" should be "nitric oxide".

We have made the change to the manuscript.

Page 8, Line 310: the word "more" should be removed.

We have made the change to the manuscript.

Page 9, Line 341: "additional" should be "additionally".

We have made the change to the manuscript.

Page 9, Line 344: the quotation mark should be removed.

We have made the change to the manuscript.

Page 9, Line 346: "mutate" should be "mutant".

We have made the change to the manuscript.

Page 9, Line 379: "Rac1 has several isoforms, but Rac1 is ubiquitously..." which Rac1 isoform?

The sentence has been clarified to say “Rac has several isoforms (Rac1, Rac1b, Rac2, and Rac3), but Rac1 is ubiquitously expressed…”

 Page 10, Line 393: "ATIR" should be "AT1R".

We have made the change to the manuscript.

Reviewer 3 Report

This is a short, mini review on GTPase or Rac1, Rho and Ras in endothelial cells. I have no objections except the following suggestions:

Major

Fig. 2 does not help.

# of references are less than 100. You started an interesting mini review and readers are about to read an interesting story, then you stopped the story (you throw away something and stopped talking: that's what readers will have as impression). In other words: one would expect:

exposition - rising - climax - falling - denouement (end): one or two components are missing in your current structure of the manuscript

Rac1 story on VEGF should be in the first. This was the most interesting or useful one.

It is less clear when and how endothelial cells (supposed not to be migratory or motile in health) transform to invasive phenotype? Any relevance/comment/linkage to MET or vice versa?

Altogether, if one categorizes the content of your current manuscript into three, I would choose the first among: 1) insufficient, 2) appropriate, 3) more than enough

Minor

line 206-224: this part confesses that your lab/your team were interested in smooth muscle cell and published the relevant research in the past. But your current review is anew and on endothelial cell. Do you have enough expertise to write a review on endothelial cells, then?

line 293: ...'statins effect...' (verb should be used) into 'affect'

Thank you.

Author Response

Fig. 2 does not help.

We agree that many of the intended readers are familiar with small GTPases and this figure does not add much for this audience.  However, given the focus on clinical implications, many readers might not be familiar with this schematic.  Furthermore, there are many abbreviations that can be confusing.  Our colleagues who have read the drafts of the manuscript all commented that they appreciated this figure.  We have elected to keep the figure in the manuscript.

# of references are less than 100. You started an interesting mini review and readers are about to read an interesting story, then you stopped the story (you throw away something and stopped talking: that's what readers will have as impression). In other words: one would expect:

exposition - rising - climax - falling - denouement (end): one or two components are missing in your current structure of the manuscript

We have focused the manuscript to deliver a more coherent message.

Rac1 story on VEGF should be in the first. This was the most interesting or useful one.

We agree with the reviewer, and we have moved the section on RhoA.

It is less clear when and how endothelial cells (supposed not to be migratory or motile in health) transform to invasive phenotype? Any relevance/comment/linkage to MET or vice versa?

We agree that endothelial cells should be neither migratory nor proliferative in health.  However, we are focused on a specific time and location where increased migration and proliferation would be desirable – namely the treated segment of blood vessel in the immediate periprocedural period. We have attempted to make this point clearer (lines 65-74 in the Introduction, lines 459-470 in the Conclusions).

Altogether, if one categorizes the content of your current manuscript into three, I would choose the first among: 1) insufficient, 2) appropriate, 3) more than enough

We have made all the recommended revisions.  With these revisions, we hope that our manuscript has moved closer to the second category.

Minor

line 206-224: this part confesses that your lab/your team were interested in smooth muscle cell and published the relevant research in the past. But your current review is anew and on endothelial cell. Do you have enough expertise to write a review on endothelial cells, then?

We study Vascular Biology, in particular the mechanisms of restenosis.  This process is driven by both vascular smooth muscle cells and endothelial cells.  We have previously published multiple manuscripts investigating the response of both cell types to iatrogenic injury.  We have expanded the manuscript to address this point and have added references 14, 15, 90, and 91.  

line 293: ...'statins effect...' (verb should be used) into 'affect'

We have made the changes to the manuscript.

Round 2

Reviewer 3 Report

I hve no more objection.